

# Predicting thunderstorm risk probability at very short time range using deep learning

Mélanie Bosc[1], Adrien Chan-Hon-Tong[2], Aurélie Bouchard[1], and Dominique Béréziat[3]

[1]DPHY, ONERA, Université Paris-Saclay, 91120, Palaiseau, France
[2]DTIS, ONERA, Université Paris-Saclay, 91120, Palaiseau, France
[3]CNRS, Sorbonne Université, 75005, Paris, France

**Correspondence:** Mélanie Bosc (melanie.bosc@onera.fr)

**Abstract.** Forecasting electrical activity within the atmosphere remains one of the most challenging predictions, especially due to the chaotic nature of thunderstorms. Lightning strikes are precisely located and occur very quickly, which makes this task particularly difficult. Additionally, these phenomena pose a significant risk to aviation, as they statistically strike each aircraft more than once per year. Over the years, several techniques have been employed for very short-term lightning forecasting

(lower than one hour and every five minutes), such as observation-based methods and, more recently, deep learning methods. Previous studies often face difficulties in accurately forecasting lightning probability, and even with AI-driven methods, it is still difficult to obtain calibrated outputs. To address this limitation, we propose a methodology that successfully predicts lightning risk using Convolutional Neural Networks (CNNs) with attention mechanisms. The network is fed with satellite observations and Numerical Weather Prediction (NWP) outputs formatted as a spatio-temporal sequence. Results show a $F_1$

score of $0.65$ for 5-minute predictions and $0.5$ for 30-minute predictions with a very low Expected Calibration Error (ECE) of less than $10\,\%$. Thanks to the well-calibrated outputs, risk probability maps can be plotted, showing areas with strong to low chances of having electrical activity.

## 1 Introduction

In the context of aeronautics, one of the main dangers that planes encounter along their routes is the presence of cumulonimbus

clouds. Indeed, these clouds are associated with strong updrafts and downdrafts, as well as the potential presence of hail and severe turbulence. In addition, these clouds are the main natural lightning generators on Earth, representing a major hazard for airplanes (Holton, 2004). Commercial aircraft are, on average, struck by lightning once a year (Uman, 1987), which requires grounding and mandatory maintenance operations to ensure that no significant damage has occurred. These operations lead to both time and financial losses for airline companies. Furthermore, when lightning strikes an aircraft, its

structure may be damaged depending on the intensity of the strike (Chemartin et al., 2012). Electronic malfunctions can also occur, and lightning-induced arcs may cause fuel tank explosions (Plumer et al., 1982; Laroche et al., 2012), potentially leading to serious accidents, as was the case with Pan Am Flight 214 from San Juan to Baltimore, which resulted in the deaths of 81 passengers (Laroche et al., 2015). Lightning strikes also pose a human risk, as they may hurt pilots and onboard people (EASA, 2018; Bouchard et al., 2024; Mäkelä et al., 2013). Beyond aeronautics, thunderstorms also threaten various sectors such as



public safety, insurance, agriculture, and even rocket launches. To this end, certain standards have been established, such as
the requirement for commercial aircraft always to remain at least 32 km away from a thunderstorm classified as dangerous
or from areas with very intense radar reflectivity signals (FAA, 2013). Other methods have also been developed, such as the
implementation of a tactical support system based on statistical analysis of thunderstorm trends, which allows up to 80 % of
lightning events to be avoided (Yoshikawa and Ushio, 2019). Finally, lightning detectors can also be directly integrated with

the radar systems or lightning mappers located in the aircraft's nose to alert pilots of lightning activity along their route (EASA,
2018; Milani, 2025).

     To enhance overall air safety, the ALBATROS project[1] from the European Union Horizon Europe was launched in 2023.
The present study aligns with one of ALBATROS's objectives, which is to develop safety risk models to predict and prevent
emerging hazards in aviation, and particularly lightning ones. Given that lightning is a dangerous phenomenon, the ability

to forecast it is therefore essential. In addition, cumulonimbus clouds, which form when moisture, instability and a triggering
mechanism are present, can become very large in the case of multicellular storm systems, whereas isolated storms typically have
a characteristic size of approximately 10 km and a duration of about 1 hour. Consequently, forecasting such punctual events is
particularly challenging due to the chaotic nature of storm systems and lightning itself, as well as their spatio-temporal scales.
Several forecasting methods have been developed to predict thunderstorms and lightning activity for lead times of less than a

few hours (Wilson, 1998), which corresponds to a nowcasting task.

     On the one hand, numerous prediction methods have been developed based on the use of one type of observation. For exam-
ple, thunderstorms can be forecast using radar data, which are then extrapolated to predict future electrical activity (Dixon and
Wiener, 1993; Johnson et al., 1998; Handwerker, 2002). Extrapolation techniques have also been developed using satellite data
alone (Zinner et al., 2008), lightning detection network data (Betz et al., 2008; Pédeboy et al., 2016), or radiosonde data (Sénési

and Thepenier, 1997). Techniques based on the fusion of heterogeneous data can also improve forecast performances through
extrapolation, as in Meyer (2010); Kober and Tafferner (2009); Burrows et al. (2005), or by using optical flow methods which
rely on satellite or ground-based lightning detection data to predict convection (Müller et al., 2022).

     On the other hand, statistical methods such as belief functions (Dezert, 2021; Bouchard et al., 2022), fuzzy logic-based
approaches (James et al., 2018), or integro-differential modeling (North et al., 2020) can be used to estimate thunderstorm threat

probability based on storm-related predictors. Other methods rely directly on Numerical Weather Prediction (NWP) models,
which simulate and forecast the convective processes responsible for the initiation and maintenance of thunderstorms (Lynn et
al., 2012; Dafys et al., 2018). The outputs of these models may be combined with ensemble techniques to produce probabilistic
weather forecasts, such as in Bouttier and Marchal (2020).

     However, extrapolation-based forecasts become unreliable beyond a 30-minute lead time due to a significant drop in per-

formance (Wilson, 1998). In parallel, statistical approaches are often highly complex, requiring a deep understanding of thun-
derstorm physics and involving high computational costs. These limitations have motivated the development of alternative
forecasting strategies.

---

[1]https://www.albatros-horizon.eu





In recent years, the advent of Artificial Intelligence (AI) has enabled significant progress in weather prediction methods. Meteorological sensors are now widespread, and a vast amount of data is available from observations (satellite-based, ship-based, airborne, ground-based platforms), as well as from reanalysis derived from NWP outputs. Neural Networks (NNs) have made it possible to leverage this data to produce long-term forecasts on a global scale that can compete with traditional NWP methods (Bodnar et al., 2024; Andrychowicz et al., 2023; Lam et al., 2023; Price et al., 2024; Couairon et al., 2024).

IA methods have shown promising results in forecasting mesoscale meteorological phenomena such as convective cells or intense convection (Pan et al., 2021; Bouget et al., 2021). In addition, various studies have demonstrated the feasibility of predicting thunderstorms or lightning activity using NN and different types of input data, such as ground-based lightning detection networks, satellite observations, radar data (Collins et al., 2016; Brodehl et al., 2022; Geng et al., 2021; Bosc et al., 2024; Zhou et al., 2020; Cintineo et al., 2022) and even NWP outputs (Creswick, 2025; Korpinen et al., 2024; Leinonen et al., 2023). These approaches have been applied successfully across a wide range of forecasting horizons, extending up to 2, 3, or even 6 hours. However, AI-based methods tend to be poorly calibrated and often rely on radar data, which limits their suitability for forecasting lightning along aircraft flight paths.

In this work, we developed an NN-based method to forecast lightning risk probability at mesoscale, every 5 minutes up to 1 hour, providing calibrated output maps without relying on radar data, which leverages spatio-temporal sequences to generate forecasts. Several AI-based architectures are already well adapted to this type of input, such as ConvLSTM (Shi et al., 2015) and PredRNN (Wang et al., 2022), making them particularly suitable for weather forecasting tasks. More recent approaches, such as attention mechanisms, have shown improvements in time series prediction (Janny et al., 2022; Yu et al., 2024; Lin et al., 2019), which motivated the use of an attention-based model, ED-DRAP (Che et al., 2022), in our work. Beyond architectural choices, the design of the loss function plays a critical role in producing well-calibrated outputs. While the issue of calibration in neural networks has been extensively studied (Guo et al., 2017; Nixon et al., 2019; Wang, 2024), it is still rarely addressed explicitly in the context of weather forecasting applications. For this reason, a specific study was carried out on the choice of the loss function to ensure output calibration and enable the generation of physically interpretable lightning forecast maps.

This paper is organized as follow. Input data are described in Sect. 2, and are then fed as spatio-temporal sequences to an adapted version of ED-DRAP, detailed in Sect. 3, to generate probabilistic risk maps of lightning activity. The outputs are successfully calibrated and are then analysed in Sect. 4, followed by a discussion in Sect. 5.

## 2  DATA

To train the prediction model, relevant data must be collected to extract features related to lightning and thunderstorms. These data are collected from satellite's sensors and outputs of NWP models over a specific area.

### 2.1  Studied area

Data is collected from different meteorological sources over a precise area to restrict the study. The selected area detailed in Figure 1 is named Continental United States (CONUS), and it was chosen due to the availability of sensors with high spatial





and temporal resolution. The area is also near the Intertropical Convergence Zone (ZCIT), which has a well-known hot and humid climate that favors the occurrence of thunderstorms (Bouchard et al., 2022; Hobbs, 1987; Virts et al., 2013).

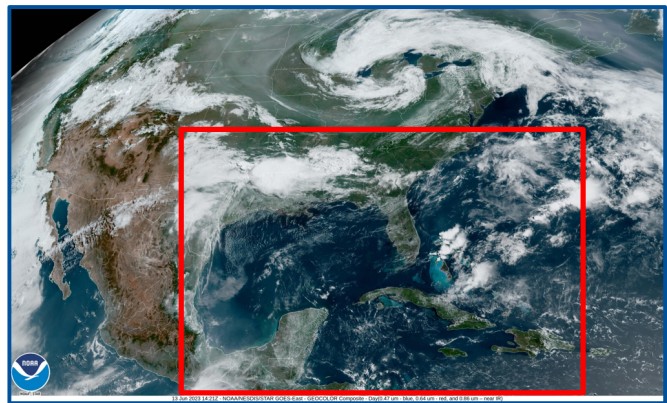

**Figure 1.** The blue rectangle represents the geographical studied area CONUS, and the red rectangle represents the final smaller chosen area centered over the Gulf of Mexico and Florida. https://www.star.nesdis.noaa.gov/GOES/conus.php?sat=G16

As CONUS area is very large, a smaller area delimited by the red rectangle centered over the Gulf of Mexico and Florida has been chosen to reduce the study's dimensions and the training computation cost. The exact coordinates of the studied area are [15 °N ; 40 °N] degrees for latitude and [100 °W ; 65 °W] degrees for longitude.

## 2.2 Satellite data

Satellite data are relevant in the context of detecting and predicting thunderstorms. The selected data come from the Geostationary Operational Environmental Satellite (GOES-R/GOES-16), which belongs to the National Oceanic and Atmospheric Administration (NOAA) and is equipped onboard with six sensors. In the case of this study, the focus is on two Earth-pointing sensors: the Advanced Baseline Imager (ABI) (Schmit et al., 2017) and the Geostationary Lightning Mapper (GLM) (Goodman et al., 2013).

The ABI sensor is a multi-channel passive imaging radiometer with a spatial resolution of 0.5 km in the visible spectrum and 2 km in the infrared. It acquires data from 16 wavelength bands and has a temporal resolution of 5 minutes over the CONUS area (Schmit et al., 2017; NOAA, 2020). From the ABI sensor, the $13^{th}$ band was selected because it is more sensitive to cloud classification. This band is located in the infrared at 10.3 $\mu$m and provides radiance data. The product selected from this sensor is the Brightness Temperature (BT), which has been derived from the radiance using Planck's law. Indeed, the BT is correlated with the presence of thunderstorms because a low BT value corresponds to a high cloud top altitude, which often indicates the presence of a cumulonimbus cloud.

Regarding the GLM, it is a space-based camera that observes electrical activity in the atmosphere with a nadir spatial resolution of 8 km and a lightning detection rate of 70-90 %. This sensor operates day and night, but performs better at night due to the contrast effect. It provides three different products every 20 seconds. The first product is the events, which correspond




to sensor pixels whose luminosity value has exceeded a threshold, indicating an electrical activity. The second product is the groups, which represent a cluster of adjacent events occurring within the same integration time, illustrating lightning and its spatial extent. The last product is the flashes that correspond to a cluster of groups occurring within a 16.5 km area and within a 330 ms duration, from which the centroid is determined. For all products, the type of lightning (cloud-to-ground or intra-cloud) cannot be determined, so all the electrical activity is taken into account (Goodman et al., 2012, 2013). From the GLM sensor, the groups were chosen because they provide larger spatial information about lightning, which is beneficial for predicting the lightning risk probability areas. As the groups were collected and transferred every 20 seconds, they were aggregated over 5-minute intervals around the ABI's acquisition time to match the temporal resolution of the ABI's data.

Both datasets were undersampled using a common mesh of 0.08° × 0.08° to achieve the same spatial resolution of approximately 8.8 km depending on the latitude and longitude coordinates. In Figure 2a representing the BT, darker pixels correspond to lower brightness temperature values, indicating higher cloud top altitudes. Additionally, in Figure 2b representing the groups, white pixels correspond to lightning activity while black pixels indicate the absence of lightning, referred to here as the background. These pictures highlight that the groups appear to be strongly spatially correlated with low BT values. If the BT is low and lightning occurs within a cloud, the cloud is likely a cumulonimbus, and this is the information that should be taken into account by a model.

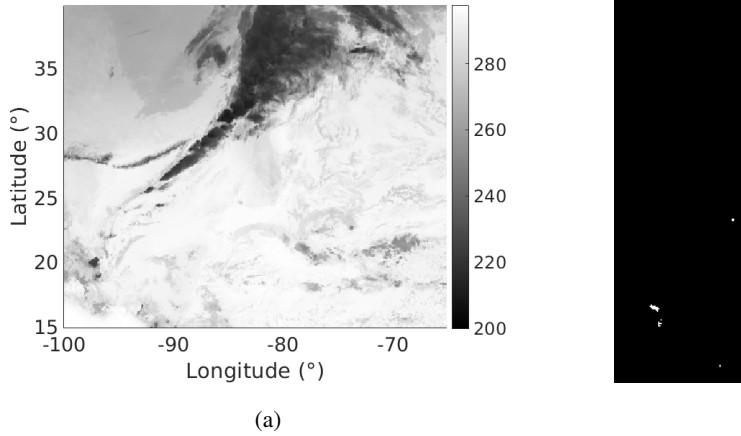
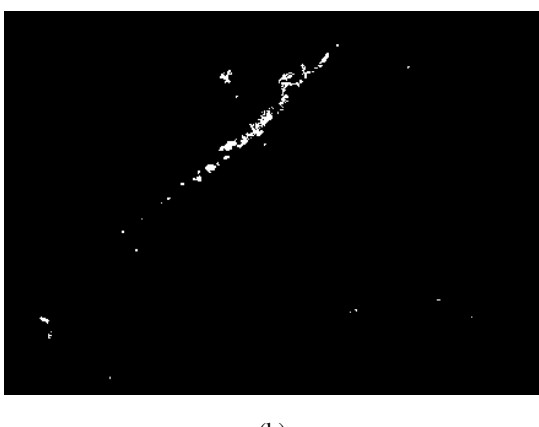

(a)                                                                                     (b)

**Figure 2.** BT image from ABI with grayscale colormap in Kelvins and darker pixels corresponding to higher BT (a). Groups product from GLM where white pixels correspond to lightning and black ones to background (b). These data are acquired on 01/13/23 at 00:06 UTC from GOES-R ABI and GLM sensors.

## 2.3 NWP Data

In addition to satellite data, NWP output data were used to enhance the information on thunderstorms. Indeed, also using these data in addition to satellite ones can help to have better forecasts (Geng et al., 2021; Leinonen et al., 2022). The Global Forecast System (GFS) (White et al., 2019) developed by the National Centers for Environmental Prediction (NCEP) was



chosen because it provides global predictions for many different meteorological parameters with a spatial resolution of 0.25°
× 0.25° and a temporal resolution of 3 hours in the archives. NWP models rely on the numerical resolution of fluid mechanics
equations, incorporating data assimilation and physical parametrizations to forecast many meteorological parameters.

In this study, two meteorological parameters related to lightning activity have been selected as inputs for the network.
The Lifted Index ($LI$) is a variable that indicates the presence of atmospheric instability, which is a key condition for the
development of cumulonimbus clouds (Haklander and Van Delden, 2003; Galway, 1956). A negative $LI$ value indicates that
the air parcel is warmer than its surroundings and can continue to rise, demonstrating the unstable nature of the atmosphere and
the presence of convection, which allows cumulonimbus clouds to form. In this study, we defined bestLI value as the lowest
$LI$ computed for different altitude levels. In Figure 3a, representing a map of bestLI value, darker pixels correspond to lower
best $LI$ values, indicating areas with a very unstable atmosphere.

The second selected parameter is derived from the Relative Humidity ($RH$) (Malardel, 2009). Here, the maximum $RH$ value
across all altitude levels (referred to as maxRH) is chosen to enable the network to identify areas with the highest concentration
of water vapor, indicating the presence of clouds in the atmosphere (Price, 2000). The maxRH is illustrated in Figure 3b, where
whiter pixels represent areas with the highest maxRH, indicating locations where clouds are very thick. Finally, the areas with
the lowest bestLI and the highest maxRH are strongly correlated with the location of lightning in Figure 2b.

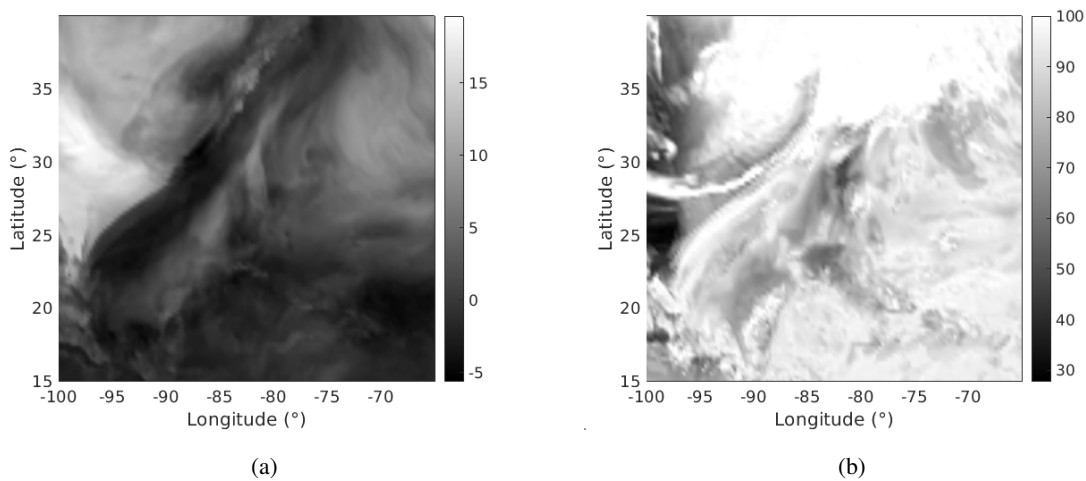

(a)                                                                        (b)

**Figure 3.** Map of bestLI (in Kelvin) with arker pixels corresponding to lower values of $LI$ so higher chances of convection (a) and map of
maxRH (in %) with darker pixels corresponding to higher maxRH so to the presence of clouds (b). These data are derived from the 00:00
UTC forecast of the 01/12/2023 18:00 UTC GFS run.

As the GFS model provides predictions at an average resolution of 25 km, oversampling was performed using Lanczos inter-
polation (Duchon, 1979) to obtain maps of the same studied area with the same spatial resolution as the brightness temperature
and groups maps. Lanczos interpolation is a method for resampling signals such as images, preserving details and reducing
artifacts, although it is more complex than simple bilinear interpolation.

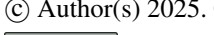



In addition, GFS provides outputs every 3 hours, which is less frequent than the 5 minutes resolution of the rest of the dataset.
To compensate, the same GFS data were reused across multiple 5-minute timesteps, introducing some redundancy. Specifically, the following configuration was adopted: 00:00 UTC forecasts were applied from 00:00 UTC to 01:30 UTC, 03:00 UTC forecasts from 01:30 UTC to 04:30 UTC, and 06:00 UTC forecasts from 04:30 UTC to 05:00 UTC.

## 2.4    Dataset creation & characteristics

Using the four datatypes (BT, groups, bestLI, and maxRH) defined in Subsect. 2.2 and 2.3, a dataset has been created using sev-
eral days in Winter over January, February, and December. The data was collected only in the morning, between 00:00 UTC and 05:00 UTC, for the years between 2020 and 2023. This period was selected to restrict the study and to observe thunderstorms at a specific time of the year. In total, data was collected for 154 days, with an average of 50 % of days with thunderstorms and lightning and 50 % of days without, to allow the neural network to adapt to every situation, even when there is no lightning activity, thus avoiding bias.
Regarding the satellite data, with images available every 5 minutes between 00:00 UTC and 05:00 UTC, there is a total of 30 images per day for each data type, amounting to 60 images per day, which corresponds to a total of 18,480 images. For the GFS output data, with outputs every 3 hours, we have 3 images per day for each data type, totalling 924 images. All of the images are about $313 \times 438$ pixels and during the training and testing phase, images are centered and cropped to reach the shape of $256 \times 256$ pixels corresponding to an area between [17.3 °N ; 37.7 °N] degrees in latitude and [93 °W ; 72 °W] degrees
in longitude. The database was randomly split once by day into 70 % for training and 30 % for testing.

## 3    MODEL

This section outlines the architecture of the proposed model. Given that all input data has been converted into images and the objective is to perform semantic segmentation to predict the location of electrical activity, Convolutional Neural Networks (CNN) have been selected for this task (LeCun et al., 1998; Long et al., 2015).

## 170    3.1    Sequencial input

The objective of this study is to predict maps indicating areas with the probability of having lightning for various timesteps. To achieve this, we deploy a semantic segmentation method, which involves generating an image or mask where each pixel obtains a confidence score of belonging to a class. By applying different thresholds over the confidence scores, pixels are classified in several bins corresponding to the probability of having electrical activity on the forecasted map.
To accomplish this task, a sequence of images was selected as the input for the neural network. To determine the optimal number of timesteps to consider, a comparative study was conducted to evaluate the performance of models using 2, 4, 6, and 8 timesteps as inputs for predicting lightning occurrences at intervals of 5, 10, 15, $\cdots$ up to 60 minutes. The study revealed that the best performance was achieved using 6 timesteps separated by 5 minutes in average, corresponding to 30 minutes for each



prediction horizon. Using 6 timesteps as input instead of 4 showed an increase of 2 % of the $F_1$ score (presented in Sect. 4.1) for predictions at 30 minutes.

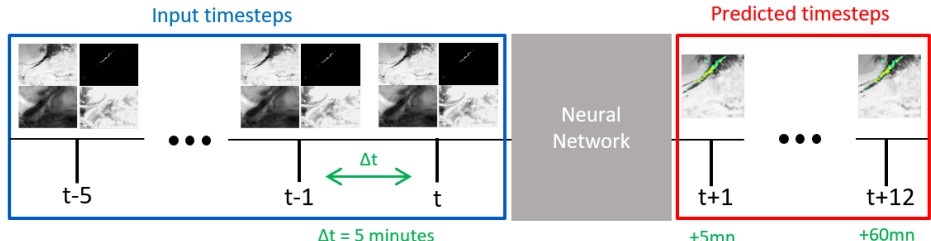

**Figure 4.** 6 input timesteps using the four datatypes separated by 5 minutes are taken to predict the lightning risk probability maps every 5 minutes ahead up to 1 hour by using a neural network.


As illustrated in Figure 4, the input tensor consists of four dimensions $[c, t, H, W]$: the first dimension $c$ represents the channels, which correspond to the number of different data types, totalling four. The second dimension $t$ corresponds to the number of selected timesteps, which is six. The last two dimensions $H$ and $W$ represent the spatial dimensions of the images, set at $256 \times 256$ pixels, as the network is trained using tiles from the input images. Using these spatio-temporal sequences, the
goal is to predict risk probability maps to have lightning at intervals of 5 minutes, up to one hour.

### 3.2 ED-DRAP

The network used in this study is named ED-DRAP (Che et al., 2022). It was first introduced for a precipitation prediction task and employs an encoder-decoder architecture with both spatial and sequential attention mechanisms. Several studies have already shown that attention mechanisms help computer vision tasks and particularly time series prediction (Guo et al., 2022;
Archambault et al., 2024; Vaswani et al., 2017).

For this study, the original ED-DRAP architecture was adapted to better fit our data and methodology. A schematic representation of the architecture is given in Figure 5. In particular, skip connections were removed after initial experiments suggested they had no positive impact on performance with our settings. In the encoder part, a first convolution is used and is then followed by encoding blocks containing 3-D convolutions, batch normalization layers, and ReLU layers. In the decoder part, some
attention modules have been added in addition to upconvolution blocks. These attention blocks are composed of a superposition of 3-D sequence attention modules and 3-D spatial attention modules and are well-described in (Che et al., 2022). Sequence attention modules called SEA are composed of an average pooling layer, followed by two convolutions and a sigmoid activation function. Then, the spatial attention module, called SPA, is composed of a 3-D convolution followed by a sigmoid layer. In both blocks, an attention mask is computed and then applied to the extracted features using a Hadamard product. The SEA
module allows the network to capture the most important time-related features, and the SPA module helps to capture the most important spatial features. Then, they are used in a residual block called SSAB, which is composed of two 3-D convolutions





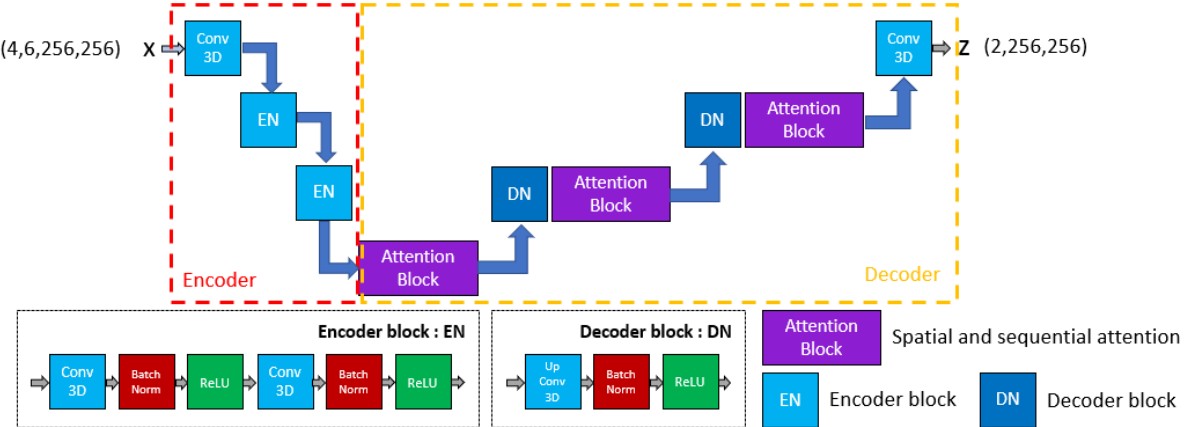

**Figure 5.** Adapted ED-DRAP architecture schematized: the encoder part is represented in red, the decoder one in yellow, and the block descriptions below. Input tensor has a shape of $[c, t, H, W] = [4, 6, 256, 256]$.

followed by the SEA module and the SPA one. In addition, the attention block defined in Figure 5 is also a residual block which uses 2 SSAB blocks and then a 3-D convolution at the end.

The model returns a tensor of shape $(2, 256, 256)$. It corresponds to a map of lightning confidence scores for both classes, lightning and non-lightning. These scores will be used to generate the probability maps shown in Sect. 4.3.

### 3.3 Calibration

As mentioned earlier, at the end of the network, a mask with values is generated. This output is then passed through a softmax layer to obtain confidence scores for each pixel, indicating the likelihood of lightning occurrence, with values ranging between 0 and 1. At this stage, machine learning studies typically stop and use these values as probabilities. However, these confidence scores cannot be interpreted as real probabilities at this step. In reality, the predicted confidence scores need to be compared to the actual frequency of events in the ground truth data to be sure that outputs are calibrated (Guo et al., 2017; Nixon et al., 2019; Wang, 2024).

To achieve this, a reliability diagram is plotted with confidence scores on the $x$-axis and the ratio of real lightning occurrences on the ground truth to the number of pixels with the corresponding confidence score on the $y$-axis. This diagram is interpreted using Figure 6a. For example, if all pixels with confidence scores between 0.35 and 0.45 are considered, an average of 40 % of these pixels should correspond to actual lightning strikes when compared to the ground truth. But if it only corresponds to 20 %, it means that the network has overestimated the probability, and if it corresponds to 60 %, it means that the network has underestimated the probability. Each reliability diagram in this article is generated from a single model trained on the full training dataset and evaluated on the entire test dataset. The selected model corresponds to the one whose performance is closest to the average over the five training runs.





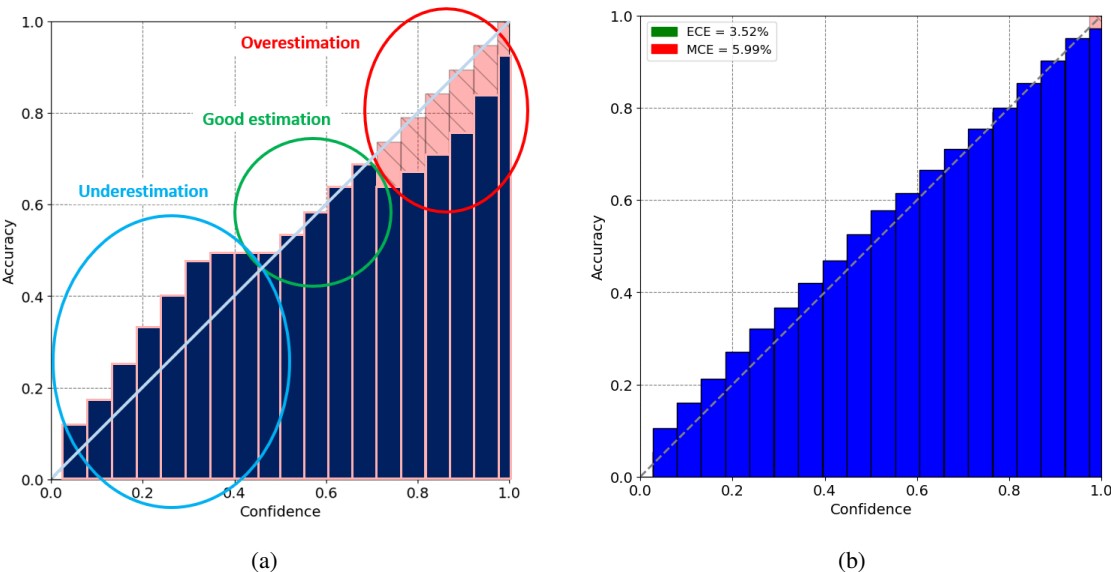

<p style="text-align:center">(a)           (b)</p>

**Figure 6.** (a) Example of reliability diagram where the light blue circle corresponds to the underestimation area, the green circle corresponds to the correct estimation area, and the red circle corresponds to the overestimation area. The perfect calibration is represented by the diagonal line. (b) Reliability diagram plotted using ED-DRAP results for 5-minute predictions averaged on the entire test set. The $x$-axis corresponds to confidence scores and the $y$-axis to accuracy (the number of real lightning over the total number of pixels classified in the bin).

Figure 6b shows a reliability diagram for 5-minute predictions using ED-DRAP, and it indicates that the network is well-calibrated because for each bin of confidence scores, the number of actual lightning corresponds to the predicted values. These results have been obtained by selecting the right input data, an adapted architecture, and by finding the best $\alpha$ coefficient in the total loss function 4, as described in Sect. 4.2.

225 In addition to the reliability diagram, calibration metrics such as the Expected Calibration Error (ECE) and the Maximum Calibration Error (MCE), defined in Eq. 1, are used to quantitatively assess how well the forecasts are calibrated.

$$ECE = \sum_{m=1}^{M} |acc(B_m) - conf(B_m)| \qquad MCE = max_{m \in 1,...,M} |acc(B_m) - conf(B_m)| \qquad (1)$$

with $M$ the total number of bins, $B_m$ the $m^{th}$ bin, $acc(B_m)$ the prediction accuracy for $B_m$ that corresponds to the total number of correct predictions over the total number of elements in $B_m$ and $conf(B_m)$ the confidence score for $B_m$.

230 These quantities are defined in Eq. 2.

$$acc(B_m) = \frac{1}{|B_m|} \sum_{i \in B_m} \mathbb{1}(\hat{y}_i = y_i) \qquad conf(B_m) = \frac{1}{|B_m|} \sum_{i \in B_m} \hat{p}_i \qquad (2)$$

with $\hat{y}_i$ the prediction, $y_i$ the ground truth and $\hat{p}_i$ the confidence for the $i^{th}$ example.





ECE measures how closely the probabilistic predictions match the actual frequencies of events on average, while MCE indicates the highest error found across all the bins. They both need to be as low as possible for a well-calibrated network's output. Calibration's results are discussed in Sect. 5.2.

### 3.4 Imbalanced problem and training

The choice of the loss function is critical for every machine learning project (Ciampiconi et al., 2021). As a matter of fact, this function will determine how well the model learns the values of its parameters and how well it can fit the data to make correct predictions.

Here, one of the biggest difficulties is that lightning pixels are less present compared to the background. This issue frequently happens and has been addressed as the imbalanced problem (He and Garcia, 2009; Wang et al., 2016). In our dataset, they represent only 1 % on average in the images of groups. Given this, using only the Cross-Entropy function as the loss function will lead to predicting only the background, as it is the majority class. This function evaluates the classification error of every individual pixel independently of their class, so if a class is overrepresented, the focus will be put on minimizing the error for this class. A solution implemented in this study is to add another term to the training loss, which is the diceloss function (Sudre et al., 2017). It can be described as follows in Eq. 3:

$$Dice = 1 - \frac{1}{2}\left(\frac{2\sum(y_0 z_0) + \epsilon}{\sum y_0 + \sum z_0 y_0 + \epsilon} + \frac{2\sum(y_1 z_1) + \epsilon}{\sum y_1 + \sum z_1 y_1 + \epsilon}\right) \qquad (3)$$

with $\epsilon = 10^{-5}$ to avoid 0-division, $y_0$ and $y_1$ the ground truths for lightning and background classes, and $z_0$ and $z_1$, the predictions for the same classes. This function calculates the Intersection over Union (IoU) for both classes, then averages them and returns the opposite. It helps the model focus on detecting more lightning because, unlike Cross-Entropy, it evaluates the percentage of each class that is correctly predicted, regardless of the class proportion in the image. Finally, the total loss function can be written as in Eq. 4:

$$loss = CrossEntropy(y, z) + \alpha Dice(y, z) \qquad (4)$$

with $z$ the prediction made by the model and $y$ the ground truth, which is the image of groups taken at the forecasted horizon. This method can be considered self-supervised because the labels are derived from one of the input data types used to train the network, and the predictions correspond to this input at a future time step. Here, the $\alpha$ coefficient has been chosen to be equal to 0.001 following a study detailed in Sect. 4.2 showing that this coefficient value gave the best calibration scores. This choice allows the model to best balance between precision and detection and to provide well-calibrated outputs.

To address the class imbalance problem in our dataset, we also tested the Focal Loss (FL) function (Lin et al., 2015) to train the network. Specifically, it aids in predicting the minority class by imposing a higher penalty on errors related to this class, unlike the cross-entropy function, which treats all classes and pixels with equal importance. FL writes as in Eq. 5:

$$FocalLoss = -(1 - p_t)^{\gamma} log(p_t) \qquad (5)$$

with $\gamma$ being the coefficient that enables the network to focus on hard or underrepresented examples, and $p_t$ representing the model's estimated probability of belonging to the lightning class. If $\gamma > 0$, the emphasis is placed on detecting hard examples.



We tested several values of $\gamma$ such as $1, 2, 3,$ and $4$. Score results using the FL have been compared to those using the total loss defined in Eq. 4. In every configuration, performance was similar or lower than when using our loss function, and the model's output was not calibrated, showing an average of 30 % of ECE and 100 % of MCE, which is significantly worse than our results.

## 4  RESULTS

### 4.1  Metrics

To evaluate the performance of our model on the task of predicting lightning probabilistic areas for different forecast horizons, various metrics have been employed. These metrics are based on quantities such as True Positive (TP), True Negative (TN), False Positive (FP) and False Negative (FN), which are the components of the confusion matrix. Here, TP corresponds to well-identified lightnings, TN corresponds to well-identified background, FP corresponds to pixels predicted as lightning instead

of background so to false alarms, and FN corresponds to missed lightnings. Using these four quantities, some metrics can be computed such as Probability Of Detection (POD) or Recall, Precision, and $F_1$ score as described in Eq. 6.

$$recall = \frac{TP}{TP+FN} \qquad precision = \frac{TP}{TP+FP} \qquad F_1 = 2\frac{recall*precision}{recall+precision} \qquad (6)$$

Here, recall corresponds to the number of well-identified lightning over the total number of real lightning, precision corresponds to the number of well-identified lightning over the total number of predicted lightning and the $F_1$ score corresponds

to the harmonic mean between recall and precision. Precision, recall, and $F_1$ score should be as high as possible for the network to achieve good prediction performance and are computed here for the positive class, so lightning. These metrics will be computed to compare the performances of different models in Sect. 5.1.

### 4.2  Choice of the $\alpha$ coefficient

To choose the right $\alpha$ coefficient in front of the dice loss function, a study of its impact on calibration has been made using

ED-DRAP to make a forecast at 30 minutes, and results can be seen on Figure 7a and 7b.

ECE and MCE values are minimal when the chosen $\alpha$ coefficient is $0.001$. Here, the test has only been plotted for a forecasted horizon of 30 minutes, but the results are the same at different forecasting horizons. Consequently, $\alpha = 0.001$ is always the best choice to have a calibrated output, so this value is kept for the entire study.

To be sure that the well-calibrated output of the network ED-DRAP is a specificity of the combination of the architecture,

the methodology, and the adapted loss function, the $\alpha$ coefficient in the loss function for the training of ConvLSTM network has also been modified. The tests are also plotted for a 30-minute forecast, and results can be observed in Figure 7.

Figure 7 highlights the fact that the best scores are achieved when $\alpha = 0.001$ for both architectures, and that ConvLSTM never reaches ECE and MCE scores lower than that of ED-DRAP. It is a strong clue that a great calibration score may be obtained by combining an optimal $\alpha$ value in the loss function and a suitable architecture.





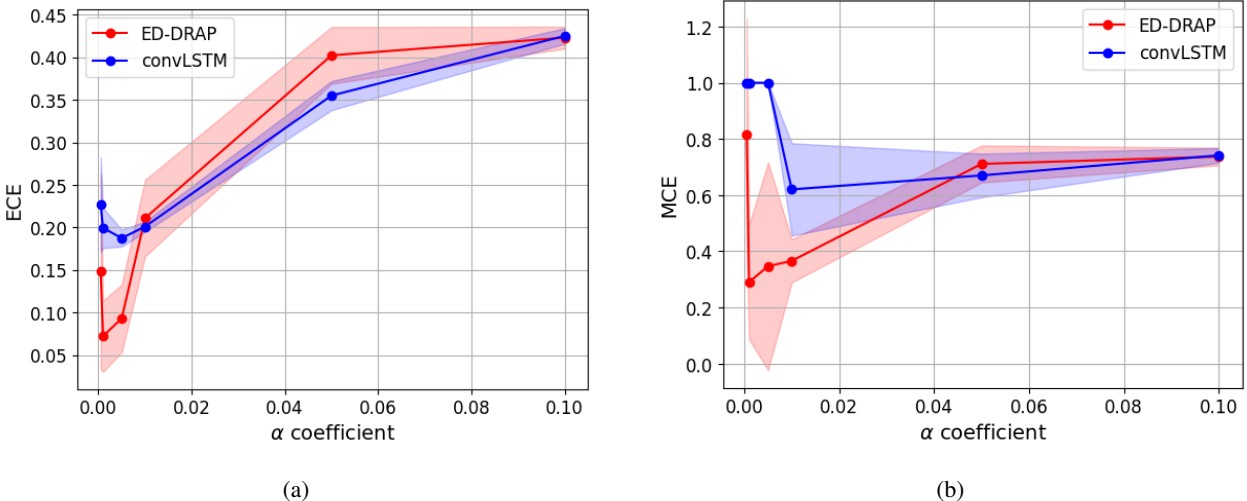

(a)          (b)

**Figure 7.** ECE (a) and MCE (b) means (solid lines) with their standard deviations shown as shaded areas, plotted against different $\alpha$ values for 30-minute forecasts using ConvLSTM and ED-DRAP networks.

## 4.3 Probability maps

Thanks to the well-calibrated outputs, physically interpretable lightning risk probability maps have been plotted. Here, the red color has been chosen for probabilities between 90 and 100 %, down to blue for probabilities around 5 %. Regions without coloration are considered to have a negligible or zero probability of being hazardous. Using this method, risk probability maps have been created, as shown in Figure 8, for different lead times and for a specific day and time in the test database.

These figures illustrate that high-risk probabilities are less frequent when forecasts are made for longer lead times, and they are also less precise, as expected. For instance, in Figure 8d, no risk probabilities exceed 65 %, indicating that the algorithm is less confident than for shorter lead times, such as in Figure 8b. To create probability maps, the output from the inference step is used. This enables the application of different probability thresholds to color-code the various lightning risk levels based on the pixel values in the output mask. Each pixel is assigned to a bin according to its confidence score, which lies within the [0, 1] interval and serves as the basis for classification.

An example of a probability map is plotted in Figure 9 with the lightning ground truth overlaid on it in order to visually see how well the real lightning fits on the prediction risk areas. This map was determined using the ED-DRAP model for predictions 30 minutes ahead on 13/01/2023 at 00:56 UTC. It shows that nearly all lightning strikes are predicted within the colored areas. In this example, only 5 % of lightning strikes are missed, and this result remains consistent across all forecast horizons, as shown in Fig. 10c.





**Figure 8.** Lightning risk probability maps for various forecast horizons obtained using ED-DRAP and the developed methodology. Forecasts for 5 minutes at 00:31 UTC (a), 20 minutes at 00:46 UTC (b), 45 minutes at 01:11 UTC (c), 60 minutes at 01:26 UTC (d) for the 13/01/2023.

## 4.4 Threshold modification

A threshold $\tau$ is used to determine whether a pixel with a given confidence score should be classified as lightning or as background at the output of the model. Typically, a threshold of $0.5$ is applied, meaning that pixels are classified as lightning if they are more likely to be so than background. However, this threshold can be adjusted depending on the application. For instance, lowering the threshold results in more pixels being classified as lightning, which increases recall but decreases precision. In this study, since the primary objective is to detect lightning activity while minimizing false alarms, the threshold




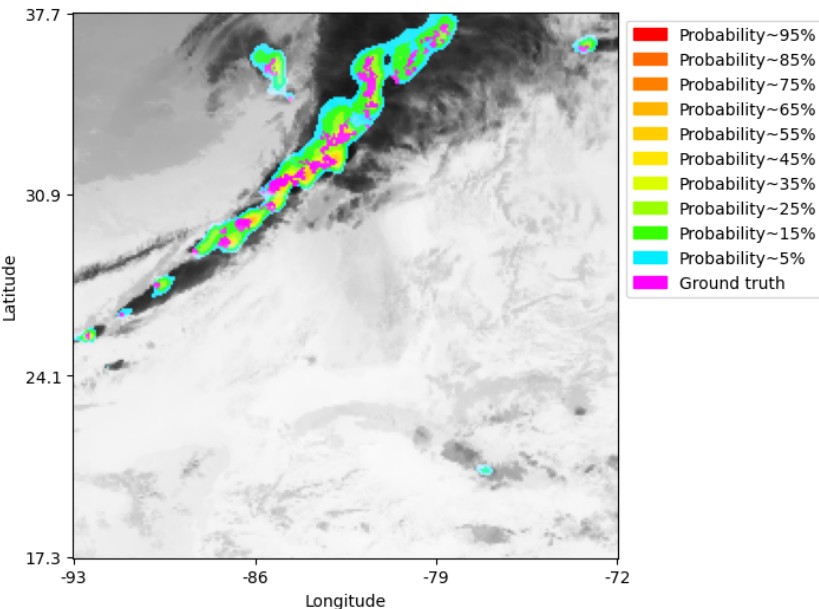

**Figure 9.** Risk probability map using ED-DRAP for 30-minute forecast horizon with ground truth plotted in magenta on 13/01/2023 at 00:56 UTC.

can be lowered. Nevertheless, the choice of threshold can be left to the user, who may prefer to prioritize detecting lightning activity or reducing false alarms. Figure 10 shows the different metrics obtained across all forecast periods for various threshold values.

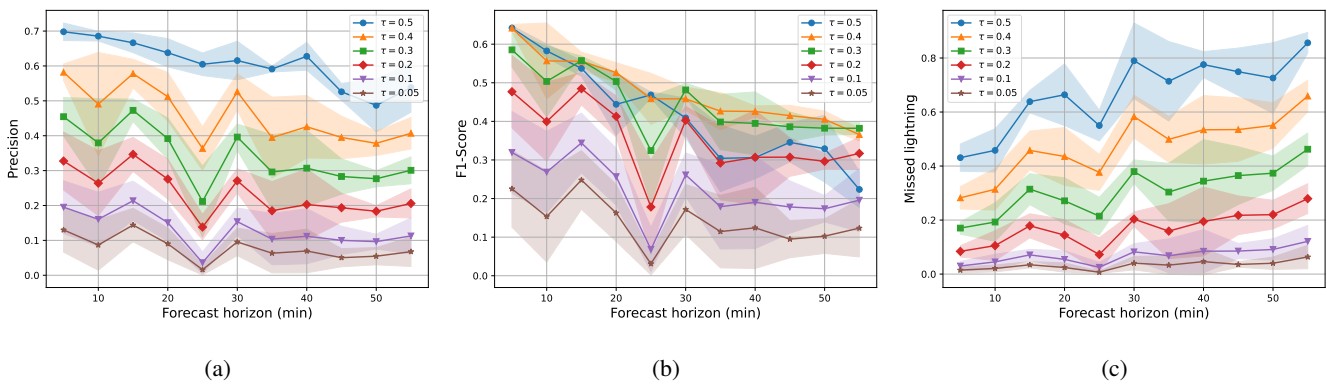

**Figure 10.** Evaluation metrics means (solid lines) with their standard deviations shown as shaded areas, plotted against several forecast horizons using different probability thresholds $\tau$. On the $x$-axis, the lead times in minutes, and on the $y$-axis, the precision (a), the $F_1$ score (b), and the missed lightning ratio (c).





POD, $F_1$ score, and the percentage of missed lightning have been plotted in Figure 10 for several lead times and thresholds. If a threshold of $0.5$ or $0.4$ is used, the $F_1$ score is maximized with a high precision score but a higher percentage of missed lightning. As the threshold decreases, precision and the $F_1$ score also decrease, but the percentage of missed lightning significantly decreases. Therefore, we selected a threshold of $0.05$ to plot blue areas on the risk probability maps to capture the maximum number of lightning strikes. Since lightning is a hazardous and punctual phenomenon, the primary objective is to

achieve a high recall while maintaining an acceptable level of precision. A suitable trade-off can be obtained by selecting a threshold of 0.2 or 0.3, which yields a recall of approximately 80 % and a precision of around 35 %. Given that the False Alarm Rate (FAR) is defined as $1 -$ precision, this corresponds to a FAR of about 65 %. Such a value remains acceptable, considering that the model aims to identify large areas at risk to issue warnings and minimize the occurrence of undetected lightning events.

## 5   Discussion

### 5.1   Scores comparison between models and forecast horizons

To ensure that ED-DRAP truly outperforms other networks, it is necessary to compare the scores obtained by testing various other spatio-temporal predictor networks using different metrics. We selected two well-known networks for comparison with ED-DRAP: ConvLSTM (Shi et al., 2015) and PredRNN (Wang et al., 2022). The ConvLSTM model used here is the vanilla version, with 2 layers of convolutional LSTM and 64 hidden dimensions. The same parameters were used for the

classical PredRNN in this study. ED-DRAP model is also compared to these networks, as well as to persistence and to a U-Net model (Ronneberger et al., 2015). Results for U-Net are not plotted here because the network only provides results for a 5-minute forecast period and is not able to forecast to longer horizons. This can be explained by the fact that it is a segmentation network, but it has not been created to catch spatio-temporal dependencies and to make multiple timesteps predictions using it. For 5-minute forecasts, U-Net achieves an $F_1$ score of $0.44$, which remains lower than those obtained by the other

spatio-temporal neural networks, as it can be seen in Figure 11.

Here, each network was trained with 100 epochs, the loss function described in Subsect. 3.4, the Adam optimizer, batches of 2, and a learning rate of $0.0001$. The computed metrics, described in Subsect. 4.1 for recall, precision, and $F_1$ score, and in Subsect. 3.3 for ECE and MCE, were calculated using a threshold of $0.5$. This means that if a pixel's confidence score is above $0.5$, it is forecasted as lightning. We chose to use this threshold first to compare scores with other methods, but this choice is

discussed in Subsect. 4.4.

A distinct model is trained for each forecast horizon, allowing predictions to be optimally adapted to each lead time. All metrics were computed over an ensemble of 5 trainings for each forecast horizon and were then averaged to ensure robustness. Train a model on our dataset takes an average of 76 minutes on a NVIDIA RTX A5000. However, the inference phase only takes 10 seconds to generate a map at a chosen lead time on a CPU. The results are presented in Figure 11.

First, for each metric, all performances decrease with the forecast horizon, which is commonly observed in meteorological predictions. Then, in Figure 11a, both PredRNN and ConvLSTM do not achieve better scores than persistence, except for ED-DRAP, which yields a higher average recall. This may be due to the fact that PredRNN and ConvLSTM lack the capacity





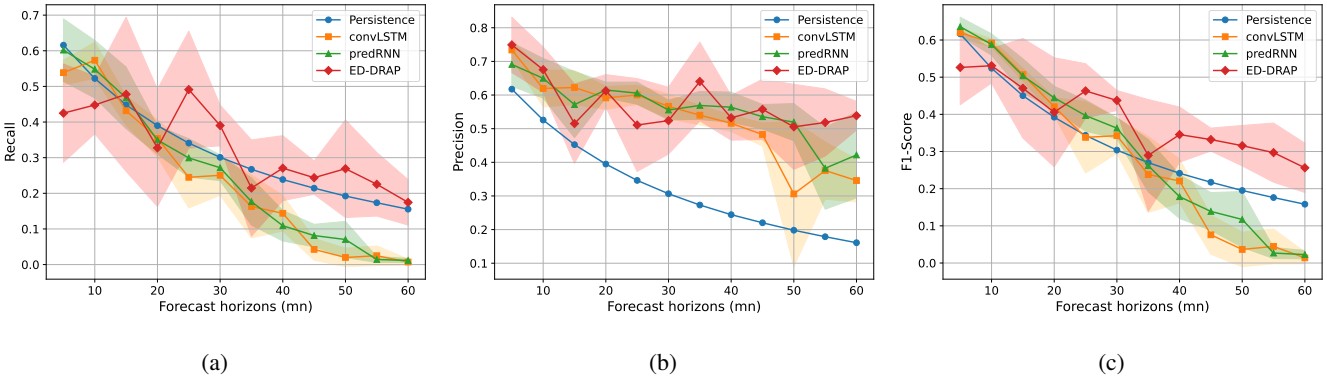

$$(a) \qquad\qquad (b) \qquad\qquad (c)$$

**Figure 11.** Recall (a), precision (b), and $F_1$ score (c) means (solid lines) with their standard deviations shown as shaded areas, plotted against several lead times for different tested models. The $x$-axis represents the lead times (minutes), and $y$-axis represents the metric value.

to capture the complex spatio-temporal correlations required to accurately predict electric activity, especially in the presence of rare and highly localized events. Additionally, in Figure 11b, persistence precision score is the lowest for each forecast period,
and all the other networks give similar results. Finally, in Figure 11c, the $F_1$ score is better using ED-DRAP, especially for longer forecast periods. This demonstrates that ED-DRAP is better suited to make predictions using our dataset and methodology because it succeeds in achieving great scores, such as a recall of $0.4$, a precision of $0.52$, and an $F_1$ score of $0.44$ for 30-minute forecasts, which are all better scores than with the other networks.

## 5.2 Calibration results comparison between models and forecast horizons

Here, calibration results are compared thanks to the ECE and the MCE metrics. First, as shown in Figures 12a and 12b, persistence cannot achieve good calibration results in this case because it merely compares one timestep to another, without involving confidence scores.

Then, the last three networks are compared over different forecast periods. The ECE and MCE are minimal when using ED-DRAP, with an average calibration error of only $10\,\%$ even for a 50-minute prediction. At this forecast period, ConvLSTM
and PredRNN have a minimum calibration error of $30\,\%$. Regarding the maximum calibration error, it reaches $100\,\%$ for every network except ED-DRAP.

To understand where the calibration errors occur, reliability diagrams can be plotted. On Figure 13, we compared the reliability diagrams of ConvLSTM, PredRNN, and ED-DRAP for a forecast period of $30\,$minutes.

ED-DRAP is the only model to achieve great calibration scores and for ConvLSTM and PredRNN, low-risk probabilities are
well-calibrated, but these networks are not well-calibrated for high confidence scores when it comes to high-risk probabilities. In fact, if all pixels with a predicted confidence score of $0.9$ are considered, $0\,\%$ of them are actual lightning strikes, leading to poor network calibration. ED-DRAP successfully gives confidence scores, which are calibrated when compared to the ground truth. All these results explain why ED-DRAP has been chosen for this electrical activity risk probability forecasting task.



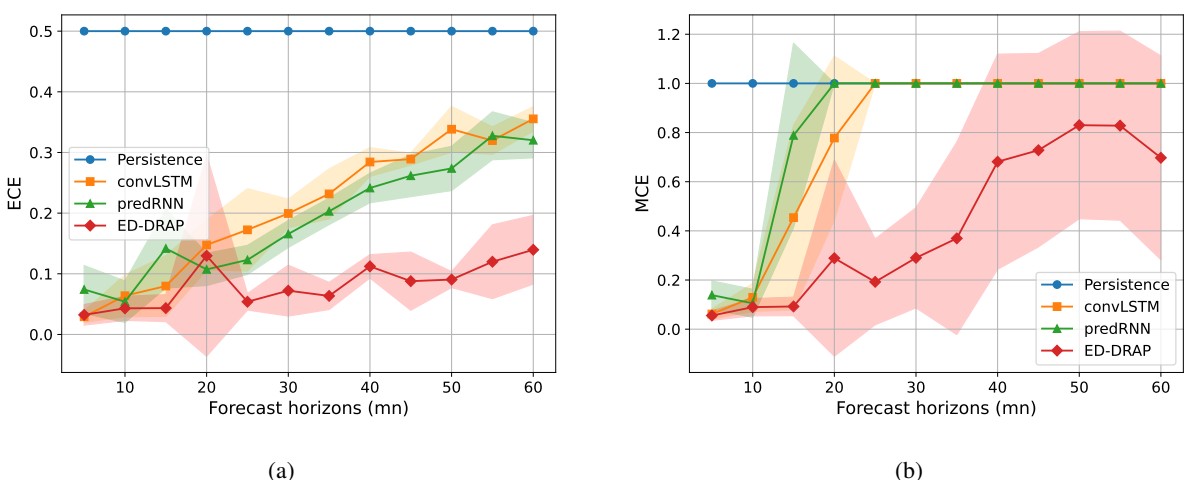

(a)           (b)

**Figure 12.** ECE (a) and MCE (b) means (solid lines) with their standard deviations shown as shaded areas, plotted against several lead times for the different tested models. On $x$-axis are the forecast periods (minutes) and on $y$-axis are the ECE and the MCE metrics.

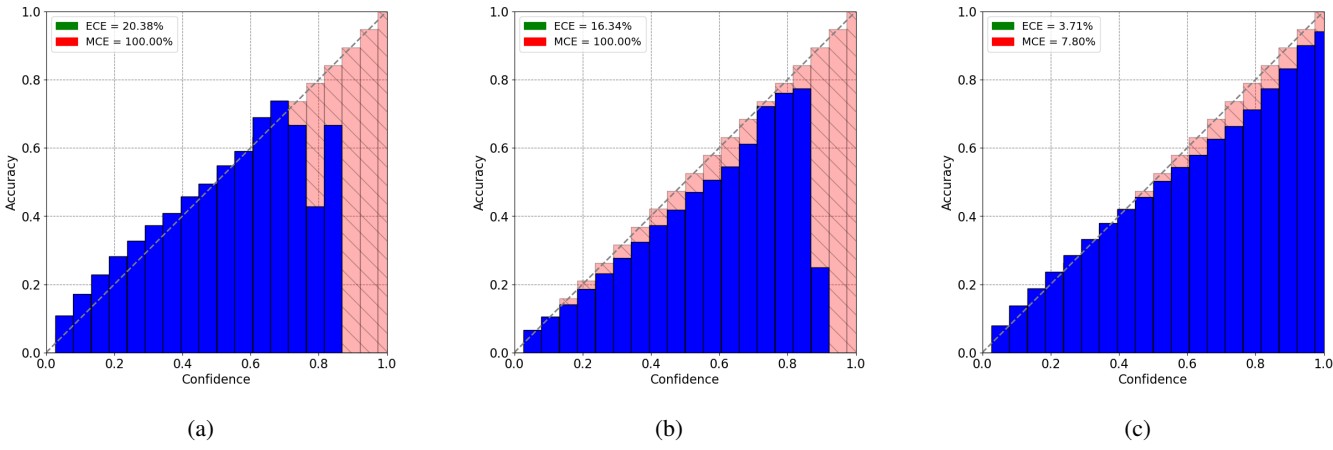

(a)           (b)           (c)

**Figure 13.** Reliability diagrams for 30-minute predictions using convLSTM (a), predRNN (b), and ED-DRAP (c). The $x$-axis represents the calibration bins and the $y$-axis represents accuracy. ECE and MCE scores are printed on the left top of the plot.

In addition, compared to the diagram in Figure 13c, the outputs are slightly less well-calibrated. Specifically, the bins do not

perfectly align with the diagonal and are positioned above it, indicating that the network tends to overestimate the presence of lightning. For instance, among the pixels with a confidence score of $0.5$, only $43\%$ actually correspond to lightning when compared to the ground truth. While the output for the 30-minute forecast remains reasonably well-calibrated, it is less so than the 5-minute forecast, which is expected as the prediction task becomes more challenging over longer timeframes.



## 6 Conclusions

In the framework of air safety enhanced by the ALBATROS project, the study has focused on developing a lightning risk probability methodology to allow airplanes to avoid dangerous stormy areas. To do that, we have created a database using meteorological parameters, such as satellite data products and outputs from Numerical Weather Prediction (NWP) models, to train neural networks for generating very short-term lightning strike risk forecast maps. The study highlights a methodology that involves using multiple timesteps as input to feed a modified version of a neural network named ED-DRAP, which is an 385 encoder-decoder model utilizing spatial and sequential attention.

The study also emphasizes the importance of selecting an appropriate loss function tailored to the problem, specifically addressing the imbalanced nature of the dataset. The article demonstrates that using the Dice Loss function can be beneficial in such imbalanced scenarios and can aid in calibrating the network's outputs.

Moreover, combining useful meteorological data with an adapted neural network and a suitable loss function can result 390 in well-calibrated network outputs. This allows for the creation of lightning activity probability risk maps using different probability thresholds and colors. These maps can be generated for various forecast periods, ranging from 5 minutes to 1 hour, while remaining well-calibrated. They can be adapted to user preferences based on the desired precision and recall scores.

Future work will focus on several directions. First, we plan to incorporate radar data into the input, as it has been shown to improve prediction performance (Leinonen et al., 2022). Second, since our current methodology requires training separate 395 models for each forecast horizon, we aim to develop a multi-horizon approach by transforming the architecture into an autoregressive model. Finally, given the promising results of diffusion models in meteorological forecasting using AI, we intend to explore their application to our task.

*Author contributions.* MB, ACHT, AB, DB conceived the study. MB conducts the experiment and writes the paper with inputs from all authors

*Competing interests.* The authors declare that they have no conflict of interest.

*Acknowledgements.* This research is co-funded by the ALBATROS project, from the European Union Horizon Europe under Grant Agreement N°101077071. We thank the NOAA National Geophysical Data Center for providing the GOES-R data and the NOAA National Centers for Environmental Prediction and National Weather Service for giving access to the GFS data. We also thank Théo Archambault for helping us with the ED-DRAP network.



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
