# Peer review of "Predicting thunderstorm risk probability at very short time range using deep learning"

_EGUsphere, 2025_

## Author Comment (AC1)

**Answer to the first review:**

**Acknowledgements:**

We thank the reviewer for their very constructive feedback and thoughtful comments. We have carefully studied the review, addressed all the points raised, and modified the manuscript accordingly.

**Main concerns:**

**First main concern:**

My main concern is the limited scope and potential lack of generalizability of the dataset and results. The data is restricted to winter mornings (00:00-05:00 UTC, December-February) from 2020-2023, covering only 154 days with a balanced split of stormy and non-stormy periods. While this controls for variability, it may not capture seasonal, diurnal, or regional differences in thunderstorm dynamics (e.g., summer afternoons or other global hotspots). The study area is narrowed to a subset of CONUS, but no sensitivity analysis is provided for other regions. A discussion on how these choices affect broader applicability, perhaps with preliminary tests on extended data, would strengthen the contribution.

**Answer to the first main concern:**

Thank you for this insightful comment. We fully agree with you, and we have conducted additional tests to demonstrate that the method also performs well across different regions, seasons, and times of day, in order to assess the model's robustness. The methodology was as follows:

We used data from another region centered over Panama, between latitudes [0, 15]°N and longitudes [100, 70]°W.

Figure 1 - New inference selected area centered over the Panama

This region does not overlap with the training and testing area, which was located between latitudes [15, 40]°N and longitudes [100, 65]°W.

Four days in August 2024 were selected during which thunderstorms occurred in this region. The data were collected and processed for the period between 12:00 UTC and 15:00 UTC. With this selection, we were able to test the robustness of the method on summer afternoons over a different region, as suggested in your comment. We used GOES-R ABI Brightness Temperature (BT), GLM groups data, and GFS best lifted index (bestLI) and maximum relative humidity (maxRH) as a spatio-temporal sequence

to infer on this new test case using our pre-trained model. Here are the obtained mean results in terms of metrics over the 4 selected dates:

Figure 2 - F1-Score plotted for forecast horizons every 10 minutes up to 1 hour using several thresholds from 0.5 to 0.05.

Visually, we also plotted a map showing the results for one of these dates (on 11 August 2025) and for a 10-mn forecast horizon.

Figure 3 - 10-mn predictions on 11 August 2025.

The network seems to successfully predict lightning activity probability across another season, time of day, and region, achieving results comparable to those obtained over the Gulf of Mexico. In future work, additional experiments will be conducted on a larger dataset, and fine-tuning will be explored for different regions.

**What we change on the paper:**

We added a new subsection in the discussion section: 5.3 – "Assessment of the robustness of the method" after line 378 which contains the following text: "To ensure that the method can capture the seasonal, diurnal, and regional differences in thunderstorm dynamics, the model's performance has been evaluated over a new region centered on Panama. This area extended over latitudes [0°, 15°N] and longitudes [100°W, 70°W] and does not overlap with the training domain. Several days in August 2024 were selected when thunderstorms occurred, focusing on the period between 12:00 and 15:00 UTC to assess the model's performance during summer afternoons. Input images were generated using

the same GOES-R satellite sensors (ABI and GLM) and outputs from the GFS model. In this region, only Full Disk data from GOES-R's sensors are available, resulting in a temporal resolution of 10 minutes. Robustness tests were conducted, and the corresponding performance metrics are presented in Figure 14a. Despite the current limited size of the Panama dataset, the method generalizes well to these new conditions, achieving F1-Scores close to those obtained over the Gulf of Mexico. Moreover, it still produces well-calibrated probabilistic maps representing the risk of electrical activity as shown in Figure 14b."

**Second concern:**

My second concern is the benchmarking and novelty assessment. The model is compared to ConvLSTM, PredRNN, persistence, and U-Net, showing superior F1 and calibration scores. However, it lacks direct comparison to recent lightning-specific DL models from the literature, such as those in Brodehl et al. (2022), Geng et al. (2021), or Leinonen et al. (2023), which also use satellite/radar data for nowcasting. While the intentional exclusion of radar data is well-justified for enhancing applicability to aircraft flight paths where radar coverage may be limited or absent, discussing how the proposed method might compare to radar-inclusive baselines would better contextualize its advantages and limitations.

**Answer to the second concern:**

Thank you for your second comment regarding the lack of direct comparison with other state-of-theart deep learning models such as Brodehl et al. (2022), Geng et al. (2021), and Leinonen et al. (2023).

First, comparing models for this type of prediction task is quite challenging, as no public benchmark currently exists. Each research group creates its own dataset, designs its network, and applies it to its data. Therefore, each model is explicitly tailored to the specific dataset used. This means that different studies employ distinct input tensors, ground truths, loss functions, and forecast horizons, making it very difficult to adapt one model to another dataset.

In Geng et al. (2021), the authors use simulated microphysical parameters, radar reflectivity, maximum vertical velocity, lightning data from the Chinese National Lightning Detection Network, and Automatic Weather Station (AWS) data — all very different from the inputs used in our approach. Their model is based on a modified ConvLSTM architecture with a separate encoder for each data type, followed by a fusion module and a decoder. While their architecture differs slightly, it operates on the same general principle as a ConvLSTM. Therefore, we would expect results similar to those presented in subsection 5.1, as we tested this type of network on our data. They obtain good POD values for the first few hours, but also a very high false alarm rate, resulting in an F1-score around 0.25 for 30-minute forecasts — nearly half of what we obtain in this study. For these reasons, we chose to only implement the two other networks in order to compare them with our method.

Regarding the study of Leinonen et al. (2023), they employ an encoder—forecaster model using Gated Recurrent Units (GRUs) to capture temporal dependencies. Their approach is conceptually close to a ConvLSTM, and their methodology is similar to ours, as they predict lightning probability every 5 minutes up to one hour ahead. However, their model is more complex and relies on a large number of input variables (39 different data types) over 6 timesteps to directly predict 12 timesteps (i.e., up to 1 hour). Since their code is available at <a href="https://github.com/MeteoSwiss/c4dl-multi">https://github.com/MeteoSwiss/c4dl-multi</a>, we implemented their ConvGRU architecture and tested it on our data using the loss functions mentioned in their article and GitHub repository. We experimented with their Weighted Cross-Entropy (WCE) and Weighted Focal Loss (WFL) using lightning weights in [0.01, 0.5, 10] and obtained the following results:

• When using the WCE, regardless of the weight, the network misses too many lightnings while trying to avoid false alarms.

 When using the WFL, regardless of the weight, the network predicts too many lightning pixels, thus detecting all lightning but generating many false alarms.

This behavior aligns with the design of the WFL, which emphasizes recall, while the WCE reduces false alarms. With further investigation, it might be possible to find a balance between these two losses, but for now, their method does not adapt well to our data.

Finally, in Brodehl et al. (2022), the authors propose a U-Net with additional residual blocks from a ResNet architecture, referred to as ResU-Net. Their method achieves good results for lightning prediction up to 180 minutes with 30-minute intervals. Although they use different data from ours (several bands from the SEVIRI sensor and LINET lightning detection network data), they also apply a search radius to mitigate class imbalance when computing metrics. They do not specify all required parameters, such as the number of timesteps or input channels, but we still implemented and tested their architecture on our dataset with our methodology. Using different loss functions, such as WCE with lightning-to-background ratios of 1/1000 and 1/100, CE combined with  $\alpha$ -Dice loss with  $\alpha$  values in [0.1, 0.05, 0.01] and also the loss function that they used in the paper (named as "Brodehl" in the table), we obtained the following results for 5-minute predictions:

| Loss function | WCE[1/1000] | WCE[1/100] | CE+0.1dice | CE+0.05dice | CE+0.01dice | Brodehl |
|---------------|-------------|------------|------------|-------------|-------------|---------|
| Precision     | 0,127       | 0,2869     | 0,2616     | 0,3484      | 0,261       | 0,3497  |
| Recall        | 0,9782      | 0,9412     | 0,9525     | 0,9111      | 0,9524      | 0,8290  |
| F1-Score      | 0,2248      | 0,4398     | 0,4101     | 0,5041      | 0,4097      | 0,4919  |
| ECE           | 0,4756      | 0,4307     | 0,4527     | 0,4109      | 0,4526      | 0,3554  |
| MCE           | 0,9013      | 0,7731     | 0,855      | 0,764       | 0,8548      | 0,5937  |

Table 1 - Metric results using ResU-Net from Brodehl et al. (2022) and different loss functions.

All these scores are lower than those achieved in our study with the ED-DRAP network and methodology. Specifically, while we obtained an F1-score of 0.65 for 5-minute predictions, the ResU-Net reached only up to 0.5. More importantly, this network failed to produce calibrated outputs, as all average ECE values exceeded 0.35 (i.e., more than 35% error), compared to less than 5% with our approach.

**What we change on the paper:**

We added one paragraph to compare our results with those of Brodehl et al. (2022) at the end of the subsection 5.1 after line 358. The paragraph is the following: "In addition to these tests, a ResU-Net model inspired by Brodehl et al. (2022) has been implemented and trained on our dataset with the loss in Equation 4, the Weighted Cross-Entropy (WCE), and the loss defined in their paper. The obtained F1-Scores range from 0.22 to 0.50 for 5-minute forecasts, which is lower than the results obtained here, and the ECE remains above 0.35, compared to less than 0.05 in this study. This shows that our methodology is better suited to produce calibrated outputs and to address the current problem."

**Minor comments:**

**First comment:**

L90-95: Clarify why the smaller area (red rectangle in Fig. 1) was chosen beyond computation cost, does it represent typical thunderstorm regimes?

**Answer:**

Thank you for this comment. The reason why we selected a smaller area was not clearly explained in the paper. In fact, this smaller region was chosen to reduce the image size, as using the entire CONUS domain (1168×834 pixels) and training the network on 512×512 tiles would have been too computationally expensive. Moreover, we aimed to design a model specifically trained on a region with a balanced ratio of land and sea, in order to later evaluate its ability to provide useful predictions in areas without ground sensors or with missing data. From a physical perspective, this region is located near the InterTropical Convergence Zone (ITCZ), which experiences intense convective activity due to the influence of the Gulf Stream.

**What we change in the paper:**

On line 90, the mention of the ITCZ was deleted. In addition, between lines 92 and 94, we added a sentence to justify the choice of the smaller area which is the following: "It has been chosen because it is near to the InterTropical Convergence Zone (ITCZ), which experiences intense convective activity due to the influence of the Gulf Stream."

**Second comment:**

Fig. 2: Add coordinate axes (latitude/longitude) to subfigure (b) to match (a) for consistency and better spatial context.

**Answer:**

Thank you for this pertinent comment. Correcting this will help the reader understand the spatial context and avoid misunderstandings.

**What we change in the paper:**

We have modified subfigure (b) in Fig. 2, and both subfigures (a) and (b) in Fig. 3, to have the exact same format and latitude/longitude axes for consistency.

**Figure 2.** BT image from ABI with grayscale colormap in Kelvins and darker pixels corresponding to higher BT (a). Groups product from GLM where white pixels correspond to lightning and black ones to background (b). These data are acquired on 01/13/23 at 00:06 UTC from GOES-R ABI and GLM sensors.

Figure 3. Map of bestLI (in Kelvin) with arker pixels corresponding to lower values of LI so higher chances of convection (a) and map of maxRH (in %) with darker pixels corresponding to higher maxRH so to the presence of clouds (b). These data are derived from the 00:00 UTC forecast of the 01/12/2023 18:00 UTC GFS run.

**Third comment:**

L164-165: The effective training/testing area is further cropped to 256x256 pixels (17.3°N–37.7°N, 93°W–72°W) from the subselected red rectangle; consider adding this cropped boundary as an inner rectangle in Fig. 1 for clarity.

**What we change in the paper:**

We modified the first figure to add the test region with a clear blue square centered over Florida and updated the legend accordingly.

Figure 1. The green rectangle represents the geographical observed area CONUS, the red rectangle the training images area and the blue square the final chosen area centered over the Gulf of Mexico and Florida. https://www.star.nesdis.noaa.gov/GOES/conus.php?sat=G16

**Fourth comment:**

L175-180: The input sequence (6 timesteps) is justified by a comparative study, but I suggest including a table or figure summarizing F1 scores for 2/4/6/8 timesteps to support this.

**Answer:**

Regarding the choice of six timesteps as input, we initially conducted a study testing different numbers of timesteps to predict the 30-minute electrical activity risk. Four trained models were evaluated on the entire test database, and the metrics were averaged. These results led us to select six timesteps as input (representing 30 minutes) because the best F1-Score was found in this case.

Table 2 - First metrics comparison for 30-minute predictions using different number of timesteps in input.

| Timesteps in input | 2      | 4      | 6      | 8      | 10     |
|--------------------|--------|--------|--------|--------|--------|
| Precision          | 0,5705 | 0,6102 | 0,6078 | 0,6339 | 0,6637 |
| Recall             | 0,3115 | 0,3096 | 0,3323 | 0,3112 | 0,2658 |
| F1-Score           | 0,4008 | 0,4057 | 0,4286 | 0,4124 | 0,3630 |

Following your comment, we performed additional experiments by retraining ten models and computing the mean results to ensure the repeatability of our findings. The obtained results are as follows:

Table 3 - Metrics comparison for 30-minute predictions using different number of timesteps in input.

| Timesteps in input | 2      | 4      | 6      | 8      | 10     |
|--------------------|--------|--------|--------|--------|--------|
| Precision          | 0,5066 | 0,4880 | 0,5350 | 0,5444 | 0,5224 |
| Recall             | 0,2398 | 0,3304 | 0,3651 | 0,2936 | 0,2144 |
| F1-Score           | 0,2920 | 0,3474 | 0,4289 | 0,3456 | 0,2566 |

These results confirmed that using six timesteps as input for 30-minute forecasts is the optimal choice.

**What we change in the paper:**

As the paper mentioned that 6 timesteps were chosen as input and identified as the best configuration for each forecast horizon, we have modified the lines between 175 and 180 and replace it with: "To accomplish this task, a sequence of images was selected as the input for the neural network. To determine the optimal number of timesteps to consider, a comparative study was conducted to evaluate the model's performance using a different number of timesteps as inputs for predicting lightning occurrences. The study showed that the best performance for 30-mn predictions was achieved using 6 input timesteps, corresponding to 30 minutes, and this configuration was therefore used for all other horizons. Using 6 timesteps instead of 2, 4, 8, or 10 resulted in an increase of at least 8% in the F1 score (as presented in Section 4.1) for 30-minute predictions."

**Fifth comment:**

L305-310: The example in Fig. 9 misses only 5% lightning, but it's not clear which threshold is used in this case.

**Answer:**

Thank you for your remark. This addition clarifies the quantitative results for the reader and makes the methodology easier to understand.

**What we change in the paper:**

On line 309, we added the threshold's value used to obtain only 5% of missed lightnings which is 0.05 in the following sentence: "In this example, only 5% of lightning strikes are missed when using a threshold of 0.05, and this result remains consistent across all forecast horizons, as shown in Fig. 10c."

---

## Author Comment (AC2)

**Answer to the second review:**

**Acknowledgements:**

We sincerely thank the reviewer for the time and care dedicated to evaluating our work. We thoroughly considered all the comments provided in this second review and revised the manuscript to address each point as clearly and comprehensively as possible.

**Concerns:**

**First one:**

1) Since the models were trained separately for each forecast horizon, there can be concerns of incoherent forecasts between different forecast horizons. The authors should provide some discussion or visualizations on how the forecasts look between different timestamps.

**Answer:**

Thank you for this comment. Indeed, ensuring coherence across predictions from models trained for different forecast horizons is not straightforward. Since each model is trained independently and therefore learns different weights, using one model for a 10-minute forecast, another for 15 minutes, and so on, could naturally lead to inconsistencies when chaining predictions for the same application.

However, only one model corresponding to a specific forecast horizon could be employed operationally. For instance, in an aviation context, air-traffic controllers or pilots may decide to rely solely on 30-minute forecasts to anticipate regions with high lightning risk. From the user's perspective, predictions remain fully consistent because they are produced by a single model dedicated to the selected horizon. Here, predictions are made using the same model but for several instants each separated by 5 minutes: 01:01 UTC, 01:06 UTC, 01:11 UTC and 01:16 UTC. An example of predicted risk maps given by the same 30-min prediction model can be seen here:

[Figure]

*Figure 1 - 30-min predictions for 01:01 UTC*

*Figure 2 - 30-min predictions for 01:06 UTC*

[Figure]

*Figure 3 - 30-min predictions for 01:11 UTC*                    *Figure 4 -30-min predictions for 01:16 UTC*

In addition, we are currently working on an autoregressive version of the model, which will use its own predictions to generate the next ones, as part of future work.

**What we change on the paper:**

On line 346, we added the following sentence: *"[…] each lead time. For this reason, in operational settings, a single model making predictions at one forecast horizon could be used. With this model, predictions are generated every 5 minutes to ensure consistency across forecasts. All metrics were […]".*

**Second one:**

2) Was the evaluation dataset fixed for the different models per forecast horizon or was the 30% chosen separately for each horizon?

**Answer:**

Thank you for your question. Maybe this point was not well explained in the manuscript. The training and evaluation datasets are kept fixed for all training runs and for every forecast horizon to ensure consistency and to clearly assess the impact of predicting at longer lead times.

**What we change on the paper:**

On line 165, the last sentence was replaced by: *"The database was randomly split by day into 70% for training and 30% for testing, and this split was kept unchanged for all models, regardless of the forecast horizon.".*

**Third one:**

3) The training / evaluation dataset seems quite small, this also shows in the results as the results are quite jumpy from one forecast horizon to another. I wonder if there was any overfitting also due to this.

**Answer:**

Thank you for this pertinent comment. As you pointed out, the results can appear somewhat jumpy, and we believe this is mainly due to the limited number of trained networks per forecast horizon (whose results are then averaged) rather than to the size of the dataset. Indeed, the variance remains relatively high because we could not train a large number of models for each horizon, as the

computational cost would have been prohibitive. In this study, we trained five different models and averaged their results to compute the metrics.

Moreover, while a larger dataset would certainly help improve overall performance, our experiments show that the model performs well over the studied area and generalizes effectively to other regions and seasons. This suggests that overfitting is not a major concern in our case.

**Fourth one:**

4) To overcome the concerns around a small training / validation dataset, it might be interesting to see if the results generalize to a different part of CONUS - likely keeping the latitude boundaries the same but shifting the longitude bounding box more to the west. If the model yields good evaluation results trained over the Gulf of Mexico but evaluated over a different region the results might be more robust.

**Answer:**

Thank you for your comment. The first reviewer also raised this point, and your observation further confirms that strengthening the robustness assessment by applying the model to another area will improve the quality of the paper. To address this concern, we conducted additional experiments to show that the method also performs well across different regions, seasons, and times of day. Specifically, we used data from another region centered over Panama, between latitudes [0°, 15°N] and longitudes [100°W, 70°W], rather than from another part of CONUS, in order to increase the level of generalization required.

[Figure]

*Figure 5 - New inference selected area centered over the Panama*

This region does not overlap with the training and testing area, which was located between latitudes [15°N, 40°N] and longitudes [100°W, 65°W].

Four days in August 2024 were selected during which thunderstorms occurred in this region. The data were collected and processed for the period between 12:00 UTC and 15:00 UTC. With this selection, we were able to test the robustness of the method on summer afternoons over a different region, as suggested in your comment. We used GOES-R ABI Brightness Temperature (BT), GLM groups data, and GFS best lifted index (bestLI) and maximum relative humidity (maxRH) as a spatio-temporal sequence to infer on this new test case using our pre-trained model. Here are the obtained mean results in terms of metrics over the 4 selected dates:

[Figure]

*Figure 6 - F1-Score plotted for forecast horizons every 10 minutes up to 1 hour using several thresholds from 0.5 to 0.05.*

Visually, we also plotted a map showing the results for one of these dates (on 11 August 2025) and for a 10-mn forecast horizon.

[Figure]

*Figure 7 - 10-mn predictions on 11 August 2025.*

The network seems to successfully predict lightning activity probability across another season, time of day, and region, achieving results comparable to those obtained over the Gulf of Mexico. In future work, additional experiments will be conducted on a larger dataset, and fine-tuning will be explored for different regions.

**What we change on the paper:**

We added a new subsection in the discussion section: 5.3 – "Assessment of the robustness of the method" after line 378 which contains the following text: *"To ensure that the method can capture the seasonal, diurnal, and regional differences in thunderstorm dynamics, the model's performance has been evaluated over a new region centered on Panama. This area extended over latitudes [0°, 15°N] and longitudes [100°W, 70°W] and does not overlap with the training domain. Several days in August 2024 were selected when thunderstorms occurred, focusing on the period between 12:00 and 15:00 UTC to assess the model's performance during summer afternoons. Input images were generated using*

*the same GOES-R satellite sensors (ABI and GLM) and outputs from the GFS model. In this region, only Full Disk data from GOES-R's sensors are available, resulting in a temporal resolution of 10 minutes. Robustness tests were conducted, and the corresponding performance metrics are presented in Figure 14a. Despite the current limited size of the Panama dataset, the method generalizes well to these new conditions, achieving F1-Scores close to those obtained over the Gulf of Mexico. Moreover, it still produces well-calibrated probabilistic maps representing the risk of electrical activity as shown in Figure 14b."*

**Fifth one:**

5) It would be good to discuss the results separated by diurnal cycles and any peaks through the day / hours of the day.

**Answer:**

We fully agree with your comment. However, at this stage, we are not able to perform this analysis because the entire dataset was selected between 00:00 UTC and 05:00 UTC, which prevents us from comparing the model's inference performance across different hours of the day. This is something we plan to address in future work. For now, we have only conducted additional tests on a few afternoon cases over Panama, which does not allow us to complete the full analysis you suggested.

**What we change on the paper:**

We have added this idea into the perspectives section on line 396: *"[...] autoregressive model. In addition, we plan to analyze the method's performance across different hours of the day, seasons, and regions using statistical evaluations. Finally, [...]".*

**Sixth one:**

6) The authors state they selected the 13th band of the ABI sensor (infrared at 10.3mu ) because it is "more sensitive to cloud classification". While this band's Brightness Temperature (BT) is correlated with high cloud tops (cumulonimbus), the argument for selecting only this single band out of 16 is not fully explored. The addition of other relevant channels (e.g., water vapor channels) could provide complementary information about the atmospheric column. The authors could at least outline any restrictions they faced in incorporating other bands.

**Answer:**

We agree with your pertinent remark. As you noted, the 13[th] band is indeed well suited for this task because it is strongly correlated with cloud-top properties, but additional bands such as bands 8, 9, and 10 related to water vapor could also have been included. For now, we chose not to add more bands in order to focus on the most informative inputs and to limit computational cost and preprocessing requirements. Moreover, water-vapor information is partly accounted for through the use of maximum relative humidity, which integrates relative humidity across all atmospheric levels. Nevertheless, we plan to investigate the impact of adding other ABI bands as well as additional data sources, such as radar observations, in future work.

**What we change on the paper:**

In the perspectives section, we revised line 394 as follows: *"First, we plan to incorporate radar data into the input, as it has been shown to improve prediction performance (Leinonen et al., 2022), and to explore the integration of additional ABI spectral bands."*

**Seventh one:**

7) The authors use of NWP data is not entirely clear with regards to which initialization / forecast time is fed as input into the model. The authors state: "Specifically, the following configuration was adopted: 00:00 UTC forecasts were applied from 00:00 UTC to 01:30 UTC, 03:00 UTC forecasts from 01:30 UTC to 04:30 UTC, and 06:00 UTC forecasts from 04:30 UTC to 05:00 UTC".

My questions are - (a) how will this work in realtime because it seems the forecasts initialized at 6:00 UTC are being applied to init times in the past? (b) How will the operational latencies of GFS impact performance?

**Answer:**

Thank you for this comment. The explanation of how the NWP data are used may indeed not have been sufficiently clear in the manuscript. In practice, NWP operational run provide forecast products every 3 or 6 hours, with lead times ranging from 0 to 96 hours. Because our network was trained on archived data, these forecasts are only available every 3 hours. For example, in our work, we used the 00:00 UTC production time with its associated forecast products at 00:00 UTC, 03:00 UTC and 06:00 UTC.

(a) For real-time operations, the use of NWP's outputs will not introduce any difficulty. Since forecasts are delivered regularly, the method will use the most recent available production time forecast products relative to the prediction interval, ensuring optimal performance.

(b) Therefore, as long as the GFS forecasts are produced normally, no latency will be introduced and the model's performance will be preserved. In the unlikely event that a forecast cycle fails (for example, the 06:00 UTC run), the model can still rely on the 00:00 UTC forecasts to provide the predictions at 09:00 UTC for example. This fallback solution remains operational but, as NWP skill naturally decreases with lead time, a slight degradation in performance would be expected.

**What we change on the paper:**

We modify the paragraph on line 150 and replace it with the following one: *"[…] redundancy. Specifically, we only used the 00:00 UTC production time. The 00:00 UTC forecast product was applied to data between 00:00 and 01:30 UTC, the 03:00 UTC one to data between 01:30 and 04:30 UTC, and the 06:00 UTC one to data between 04:30 and 05:00 UTC."*

**Eighth one:**

8) In Figure 11. it would be more useful to have a PR curve for a few forecast forizons instead of two different figures for Precision and Recall and on the curve the impact of choosing different thresholds can be plotted. That would make it much more easier to understand the tradeoff.

**Answer:**

We understand your concern so we have plotted the PR curves for different forecast horizons using ED-DRAP.

[Figure]

*Figure 8 - PR curves for 5, 30 and 55-min predictions*

As a matter of fact, this helps better understand the tradeoff between precision and recall and how to choose the appropriate threshold for each application. These new plots show that ED-DRAP gives results very close to those of convLSTM and predRNN for short forecast horizons, but clearly performs better for longer ones.

**What we change on the paper:**

As you suggested, we replaced the precision and recall curves for each forecast horizon and each tested model in Figure 11 with the corresponding PR curves for the 30-min and 55-min lead times. We also updated the legend, which now reads: *"Precision–Recall curves for 30-min lead time (a) and 55-min lead time (b) for the different tested models and several thresholds. Panel (c) shows the evolution of the F1 score, with standard deviation represented as shaded areas, across all lead times and all tested models."*.

A sentence was added on line 326: *"[…] of around 35%. The balance between recall and precision is evaluated in Figures 11a and 11b for 30 and 55-min predictions. Given that […]"* This addition guides the reader toward the new figures and helps clarify how the choice of threshold influences the trade-off between precision and recall.

In addition, we modified the sentence on lines 351-355 by the following one: *"Then, in Figures 11a and 11b, both PredRNN and ConvLSTM fail to outperform ED-DRAP, which achieves a higher overall precision–recall balance across all thresholds. Moreover, this advantage of ED-DRAP increases with the forecast horizon. This improvement may be due to the fact that PredRNN and ConvLSTM have limited capacity to capture the complex spatio-temporal correlations necessary for accurately predicting electrical activity, particularly for rare and highly localized events."*

Finally, we chose to keep the F1 score curve, as it provides essential insight into how performance evolves across the full range of lead times.

**Ninth one:**

9) Authors state that they use 0.05 threshold to plot the risk probability map since they want high recall but that can lead to a very low precision. I think a more robust explanation of chosen thresholds and their impact on metrics should be discussed.

**Answer:**

Thank you for your comment. Indeed, since the network outputs are calibrated, the plotted maps are directly interpretable. In particular, the chosen threshold corresponds to the expected percentage of missed lightning: for instance, a threshold of 0.3 would miss 30% of lightning events but provide higher precision than a threshold of 0.05.

**What we change on the paper:**

As suggested, we modified the sentence on lines 315–316 to better explain the impact of the threshold choice on the metrics: *"For instance, a high threshold results in more missed lightning but higher precision, whereas a low threshold allows detecting more lightning with lower precision. On the maps, since the network gives well calibrated outputs, the threshold directly corresponds to the expected percentage of missed lightning."*

In addition, on lines 323–324, we revised the sentence as follows: *"Therefore, we selected a threshold of 0.05 to plot the blue areas on the risk probability map, where only 5% of lightning is missed, to illustrate the network's ability to detect a high number of lightning events."*

**Tenth one:**

10) In Figure 11(a) and (b) the result for precision and recall jumps quite a bit across different horizons and sometimes lower , sometimes higher than other models. It;s actually unclear if the model truly performs better than others. In 11(c) the ED-DRAP model actually performs worse than others for first 30 mins and then better. I think it would help to report more metrics here and better understand the performance at earlier horizons across the different baselines. Maybe visualize the probability maps for the different models.

**Answer:**

Thank you for your insightful comment. As you noted, the ED-DRAP network can sometimes yield lower metric scores compared to other state-of-the-art networks. However, the key advantage of ED-DRAP is its ability to provide calibrated outputs while remaining competitive, and even superior, for longer forecast horizons, as illustrated in Figure 12. Moreover, thanks to its attention mechanisms, the network is better able to predict over extended horizons than the other models. As suggested, we have added new metrics including F0.5 and F2 scores and missed lightning rate for the four models across all forecast horizons to enable a more comprehensive comparison.

[Figure]

*Figure 9 - F0.5 Score, F2 Score and missed lightnings using the different models for each forecast horizon*

The F0.5 score corresponds to a F-score where the precision gets the priority, and we see that ED-DRAP give far better results than the other models. In addition, the F2 Score that gives priority to the recall shows similar curves than for the F1 Score and on average, ED-DRAP is better than the other models, especially after 20-minute lead time. Lastly, the missed plot also shows that on average, ED-DRAP misses less lightnings than the other networks especially after 20 minutes lead time.

We also included probability maps from each model for 30-minute predictions to help visualization.

[Figure]

*Figure 10 - 30-mn predictions using persistence for 00:31 UTC.*   *Figure 11 - 30-mn predictions using convLSTM for 00:31 UTC.*

[Figure]

*Figure 12 - 30-mn predictions using predRNN for 00:31 UTC.*    *Figure 13 - 30-mn predictions using ED-DRAP for 00:31 UTC.*

As the convLSTM, persistence and predNN outputs are not calibrated, the maps cannot be interpreted as probabilities of having electrical activity directly. The only one that correspond to real probabilities is the one using ED-DRAP which also shows better results visually (less false alarms and high recall).